# Plant Functional Dispersion, Vulnerability and Originality Increase Arthropod Functions from a Protected Mountain Mediterranean Area in Spring

**DOI:** 10.3390/plants12040889

**Published:** 2023-02-16

**Authors:** Bruno Calheiros-Nogueira, Carlos Aguiar, María Villa

**Affiliations:** 1Centro de Investigação de Montanha (CIMO), Instituto Politécnico de Bragança, Campus de Santa Apolónia, 5300-253 Bragança, Portugal; 2Laboratório para a Sustentabilidade e Tecnologia em Regiões de Montanha, Instituto Politécnico de Bragança, Campus de Santa Apolónia, 5300-253 Bragança, Portugal

**Keywords:** bottom-up interactions, plant–arthropod interactions, ecosystem services, trait-based ecology, functional groups, Natural Park of Montesinho, biodiversity conservation

## Abstract

Plant diversity often contributes to the shape of arthropod communities, which in turn supply important ecosystem services. However, the current biodiversity loss scenario, particularly worrying for arthropods, constitutes a threat for sustainability. From a trait-based ecology approach, our goal was to evaluate the bottom-up relationships to obtain a better understanding of the conservation of the arthropod function within the ecosystem. Specifically, we aim: (i) to describe the plant taxonomic and functional diversity in spring within relevant habitats of a natural protected area from the Mediterranean basin; and (ii) to evaluate the response of the arthropod functional community to plants. Plants and arthropods were sampled and identified, taxonomic and functional indices calculated, and the plant–arthropod relationships analyzed. Generally, oak forests and scrublands showed a higher plant functional diversity while the plant taxonomic richness was higher in grasslands and chestnut orchards. The abundance of arthropod functional groups increased with the plant taxonomic diversity, functional dispersion, vulnerability and originality, suggesting that single traits (e.g., flower shape or color) may be more relevant for the arthropod function. Results indicate the functional vulnerability of seminatural habitats, the relevance of grasslands and chestnut orchards for arthropod functions and pave the way for further studies about plant–arthropod interactions from a trait-based ecology approach.

## 1. Introduction

Plants provide arthropods with key resources such as food (e.g., pollen, nectar, leaves, wood, prey/host), shelter, oviposition and mating places [1], affecting arthropods’ longevity, reproduction and dispersion [2]. In turn, arthropods are responsible for several ecosystem functions as well as multiple and indispensable ecosystem services including provisioning, regulating, supporting and cultural services [3]. Nevertheless, arthropods have been historically neglected from the research [4], remaining at around 80% of the biodiversity to be described [5]. In addition, terrestrial arthropods are declining, in some cases, drastically [4], meaning that ecosystem functions and services are being lost with unpredictable consequences. The limited knowledge about arthropod biodiversity, the biodiversity decline and the threat of climate change, highlight the extreme urgency for understanding the key factors which globally determine the arthropod diversity and their functions in order to establish conservation recommendations for arthropods and their mediated functions. 

Biodiversity can be described using unique taxonomic entities (e.g., species) and measured with taxonomic indexes (e.g., species richness) [6]. The plant taxonomic diversity has often been demonstrated to enhance the arthropod community [7,8,9,10]. Alternatively, the trait-based ecology focuses on some phenotypic traits, or functional traits, holding potential functions within the ecosystem [6]. Functional traits are morphological, physiological, phenological or behavioral characteristics shared by a group of organisms, which can be quantified and ultimately linked to their functions in the ecosystem [1,11,12]. In the case of plant–arthropod interactions, plant functional traits may provide important functions for arthropods, e.g., (i) specific resources (pollen, nectar, leaves, litter for detritivores, or prey/hosts for predators and parasitoids) for the completion of arthropod life cycles; (ii) accessibility to resources; (iii) the resource amount/nutritive quality; (iv) the resource attractivity (color, odor, etc.); (v) the resource phenology (i.e., resources match the arthropod requisites); (vi) the shelter for aestivation and hibernation, sites for oviposition or attachment points for spider webs (through vegetation architecture) [1].

Protected areas represent important living laboratories to study and understand natural ecological processes in situ [13]. In the northeast of Portugal, the Natural Park of Montesinho (PNM) is a natural mountain-protected area of 74,000 km^2^, where grasslands (composed by grasses and forbs and used for livestock feeding), chestnut orchards of *Castanea sativa* Mill (the most important crop in the region), scrublands and oak (*Quercus* spp.) forests cover most of the landscape [14]. The knowledge about arthropod biodiversity and the arthropod community in the PNM is very limited. Most of the studies in the area are focused on Orthoptera, Lepidoptera and Odonata (the preferred groups among conservationists) or on the arthropod community in chestnut orchards (due to their regional economic importance) [15,16,17,18]. As far as we know, studies about plant–arthropod interactions in the area do not exist.

In this work, the goal was to evaluate the bottom-up relationships between the plant and the arthropod communities within four relevant habitats of the PNM (grasslands; chestnut orchards; oak forests—dominated by *Quercus rotundifolia* Lam.; and scrublands—dominated by *Erica* sp., *Cistus* sp., *Cytisus* sp.). We hypothesized that different habitats host different arthropod communities and that higher plant taxonomic and functional diversity increase the abundance of arthropod functional groups. For that specifically, we aimed: (i) to describe the plant community (taxonomic and functional diversity) within the main habitats of the PNM; (ii) to describe the ground and vegetation arthropod functional communities in the study area; and (iii) to analyze the bottom-up relationships between the plant taxonomic and functional diversity and the arthropod functional diversity.

## 2. Results

### 2.1. Plant Community Composition

A total of 153 species belonging to 37 families (110 species, 34 families in May and 113 species, 32 families in June) of plants were identified. For detailed ground coverage (%) per plant species see Appendix A. Some examples of the families/species with the highest coverage were: (i) in grasslands, Poaceae (e.g., *Bromus* spp., *Holcus lanatus* L., *Lolium rigidum* subsp. *rigidum* Gaudin), Fabaceae (e.g., *Trifolium* spp.), Plantaginaceae (e.g., *Plantago lanceolata* L.), Orobanchaceae (*Rhinanthus minor* L.), Asteraceae (*Anthemis arvensis* L.) and Ranunculaceae (*Ranunculus bulbosus* L.); (ii) in the chestnut orchard, Poaceae (e.g., *Anthoxanthum amarum* Brot., *Vulpia* spp., *L. rigidum*, *Bromus* spp., *Cynosurus cristatus* L., *Aegilops triuncialis* L.), Fabaceae (e.g., *Trifolium* spp., *Vicia* spp., *Ornithopus* spp.), Asteraceae (e.g., *Anthemis arvensis* L., *Hedypnois cretica* (L.) Dum.-Courset, *Bellis perennis* L.), Rubiaceae (*Sherardia arvensis* L.) and Brassicaceae (*Bunias erucago* L.); (iii) in the scrubland, Ericaceae (e.g., *Erica australis* L.), Cistaceae (e.g., *Cistus* spp.), Fabaceae (e.g., *Pterospartum tridentatum* (L.) Willk., *Cytisus* spp.), Poaceae (*Bromus* spp.) and Fagaceae (*Quercus* spp.); (iv) in the oak forest, Fagaceae (e.g., *Quercus rotundifolia* Lam.), Cistaceae (e.g., *Cistus* spp., *Helianthemum aegyptiacum* (L.) Mill.), Fabaceae (e.g., *Genista* spp., *Cytisus* spp.), Poaceae (e.g., *Anthoxanthum aristatum* subsp. *aristatum* Boiss., *Avena barbata* Link.), Rosaceae (*Rosa* sp.) and Oleaeceae (*Fraxinus angustifolia* Vahl).

### 2.2. Plant Functional Diversity in the Multidimensional Space

The construction of the multidimensional functional space (PCoA) indicated that in May the flower shape, the color and the leaf N composition significantly contributed to all the PC of the functional space. The leaf P contributed to PC 1, 2 and 3. The plant height and leaf consistency contributed to PC 3 and 4 and the inflorescence area to PC 1, 3 and 4 (Appendix A). In June, the flower shape and the leaf N significantly contributed to all PC of the functional space. The flower color, the inflorescence area and the plant height contributed to the PC1, PC2 and PC4. The leaf P, the leaf consistency and the phenological state contributed to the PC 2 and 3, PC3 and 4 and PC 1, respectively (Appendix A).

### 2.3. Response of the Plant Taxonomic Diversity to Habitat and Month

In relation to the response of the plant taxonomic indexes to habitat and month, the plant taxonomic richness was higher in chestnuts than in scrublands in May, while in June it was not significantly different among habitats. The plant taxonomic Shannon Diversity Index (SDI) was not significantly different among habitats neither for May nor June (Figure 1, Appendix A).

### 2.4. Response of Plant Functional Diversity to Habitat and Month

Regarding the response of the plant functional diversity indexes to the habitat and month (Figure 2, Appendix A), for some indexes the seminatural habitats showed a tendency for higher values: the functional dispersion (in June) and the functional specialization (in both months) were higher in scrublands and oak forests than in grasslands and chestnut orchards. In addition, the functional vulnerability was higher in scrublands and oak forests than in chestnut orchards (in May) and the functional evenness was higher in oak forests than in chestnut orchards in both months. On the other hand, grasslands and chestnut orchards showed general higher values for the functional redundancy (higher in grasslands and chestnut orchards than in scrublands and oak forests in May and higher in chestnut orchards than in scrublands and oak forests in June).

The functional originality was higher in scrublands than in oak forests in May and the functional richness, divergence and the number of functional entities did not show significant differences among habitats or moths.

### 2.5. Arthropods Functional Community

A total of 12,081 arthropods were captured between May and June.

In the ground 6937 individuals were captured (2843 in grasslands, 1477 in chestnut orchards, 1384 in scrublands and 1233 in oak forests).

In the vegetation 6144 individuals were captured (3256 in grasslands, 1552 in chestnut orchards, 583 in scrublands and 753 in oak forests).

Predators and omnivorous were the most represented functional groups in the ground while phytophagous and predators were the most abundant groups in the vegetation. Detritivores were only present in the ground (Figure 3) and mainly captured in grasslands in June.

The most represented arthropods were Araneae and Coleoptera for predators; Coleoptera and Hemiptera for phytophagous; Diptera, Lepidoptera and Coleoptera for pollinators; Formicidae, Isopoda and Dermaptera for omnivorous; and Coleoptera and Gryllidae for detritivores (Figure 4). All parasitoids were Hymenoptera.

### 2.6. Response of Functional Groups (Abundance) to Functional and Taxonomic Diversity of Plants

Predators, omnivorous, pollinators and parasitoids were more abundant in the soil than in the vegetation. Phytophagous arthropods were more abundant in the vegetation. Predators and detritivores were more abundant in May than in June and pollinators and parasitoids were more abundant in June. The taxonomic plant richness increased the abundance of predators, phytophagous, omnivorous and detritivores while the functional plant richness reduced the abundance of predators, phytophagous and detritivores. Predators and phytophagous arthropods (and tendentially pollinators) increased with the plant functional dispersion, vulnerability (in the case of predators) and originality (in the case of phytophagous). Phytophagous and pollinators (and tendentially parasitoids) were reduced with the functional evenness (Table 1). All the models were validated by the validation test (except the pollinator model which showed some misfit).

## 3. Discussion

### 3.1. Plant Community

The most intervened habitats (chestnut orchards) showed a higher plant species richness in May than the seminatural habitats (scrublands). However, seminatural habitats (scrublands and oak forest) showed a general higher plant functional diversity for most functional indexes—functional dispersion, evenness, originality and specialization—than the most humanized habitats (grasslands and chestnuts) (Figure 1 and Figure 2). In addition, in seminatural habitats the plant functional redundance was lower and the vulnerability higher than in grasslands and chestnut orchards (Figure 2) (i.e., functional diversity variation was lower), meaning that the losses of plant species are more likely to reduce the plant functional diversity in seminatural habitats than in grasslands and chestnut orchards [19]. These results indicate the importance of intervened habitats (grasslands and chestnut orchards) for plant taxonomic diversity and seminatural habitats (scrublands and oak forests) for plant functional diversity in spring.

In this study, the PCoA showed the relationships among the plant traits within the plant community. Results indicated that the flower shape, the flower color and the leaf N composition were relevant traits to characterize the plant functional diversity in the PNM (i.e., they significantly contributed to the trait variability within the plant community). Other important traits were the leaf P composition, the plant height, the leave consistency, and the inflorescence area (Appendix A show the significant contribution of each trait to each PCoA axis in May and June). The variability of these traits across the plant community indicated them as suitable candidates to investigate arthropod–plant interactions in further research in the PNM (i.e., a plant trait generally important for arthropods can be selected, but if this trait does not vary across the plant community, it will be meaningless for the arthropod community). We recall that leaf N and P composition were extracted from the literature, therefore this result must be considered as exploratory and further research must be performed for verification.

### 3.2. Bottom-Up Relationships between Plants and Arthropods

Generally, the abundance of arthropod functional groups (predators, phytophagous, omnivorous and detritivores) (Table 1) increased with the plant taxonomic richness (higher in chestnut orchards) (Figure 1). This is in agreement with several studies which found positive effects of plant taxonomic richness, diversity and complexity on arthropod communities [20,21,22] or functional groups such as predators [9,10] or phytophagous arthropods [1,7,9,23,24]. In the study region, despite the human intervention, a general low intensity/traditional management characterizes the grasslands and chestnut orchards. Different cultures such as grasslands, rainfed permanent cultures (e.g., chestnut orchards) and cover crops are subsidized by the Portuguese administration (*Portaria* nº 50/2015). The financial support is based on the positive environmental implications of these cultures and practices, which target mainly the soil conservation. Although not addressed in this study, these conservation measures may be also positively affecting the arthropod function.

In relation to the effects of the plant functional diversity on arthropods, the plant functional dispersion (predators, phytophagous and tendentially pollinators), vulnerability (predators) and originality (phytophagous) (Table 1) (generally higher in seminatural habitats) (Figure 2) positively influenced the abundance of the arthropods functional groups, while the plant functional richness (in the case of predators, phytophagous and detritivores) and evenness (phytophagous, pollinators and tendentially parasitoids) decreased it (Table 1). These results suggest that the more different and singular the plant traits are, the higher the abundance of arthropod functional groups, and therefore unique traits (and not the whole functional diversity) may have greater importance for the functional diversity of arthropods. In this sense, Gagic et al. [25] suggested that the presence of specialist arthropods may explain a larger effect of single traits than of multi-trait indices. Further research should address (i) single or unique traits effects, with particular attention to the above-mentioned traits (e.g., flower shape, the color or the leaf N composition), on arthropod functions and (ii) the effects of the functional biodiversity considering the degree of the arthropod specialization. Single traits were previously found to influence arthropod functional groups. For example, ground predators were positively correlated with the leaf litter moisture and aerial predators with the understory height, ground omnivores with the tree density, aerial omnivores with the leaf litter biomass or detritivores with the leaf litter moisture [26]. In the case of pollinators, the corolla accessibility should match pollinator mouth parts to enable the nectar consumption [27,28]. In our study, pollinators may be favored by specific plant traits in chestnut orchards—the habitat with the lowest evenness (Figure 2c,d)—but also with an elevated abundance of Fabaceae (Appendix A, Appendix A), which are visited by an entomophilous specialist. Accordingly, Fornoff et al. [29] found that the single trait diversity, particularly of flower reflectance and morphology, were important predictors of pollinator visitation, while functional diversity did not affect pollinator species richness and reduced visitation frequency. Moreover, in a long-term experiment, Scherber et al. [30] found that phytophagous species responded more strongly to changes in the plant diversity than did carnivores or omnivores, dampening with the increasing trophic level, and that the effect of plant diversity reduced with the increasing degree of omnivory, highlighting the necessity of longer experiments.

In this study, predators, omnivorous, parasitoids and pollinators were generally more abundant in the soil, while phytophagous arthropodswere more abundant in the vegetation (Table 1). Accordingly, Albacete et al. [26] found phytophagous species positively related to the understory cover in chestnut woodlands. The fact that more pollinators and parasitoids were captured in the soil than in the vegetation is surprising because these groups are mostly composed of flying arthropods and their activity occurs mainly aboveground. This result may be due to the method of capture, since for the soil arthropods pitfalls remained seven days in the field and flying insects may fall into the trap by being attracted to the water, while for the vegetation some flying insects may escape from the sweeping net. In addition, for some functional groups different capture methods may have been more reliable (e.g., visual counts for pollinators visiting flowers, since pollinators in pitfalls can draw on pitfall traps while searching for flowering plants). Predators and detritivores were more abundant in May than in June while pollinators and parasitoids were more abundant in June. This may be related to a higher resource amount for each group in each month.

This research involves several limitations which should also be considered: The trait selection constitutes a critical and not easy step in the trait-based ecology [6] and here, because it was the first attempt to establish arthropod–plant interactions in the region, the selection of plant traits and their categorization was performed at a coarse level and might be inaccurate. For example, theoretical amounts of N and P were used [31,32,33,34,35], but the composition in the growing conditions of the study region must vary (at least) to some extent. Moreover, the human spectra visual perception was used to record the flower color, but the perception of different arthropod species performs at variable UV reflectance [29]. Important plant traits for arthropods such as the resource quality or quantity, resource types (e.g., stems, roots, sap, nectar pollen), olfactory and gustatory signals, chemical attractant or repellents, leaf structure traits (e.g., cuticula thickness, lignin content, leaf area, etc.) and other structural properties of plants (e.g., grown form, density and orientation of branches and leaves, surface features, etc.) or of vegetation (e.g., density or litter quantity) were not considered [1]. Despite the potential trait inappropriateness, Gagic et al. [25] found that “most multi-trait functional diversity indices were weakly affected by trait choice, and while excluding traits worsen explanatory power in some cases, it rarely increased it”.Different arthropod taxonomic groups, even of the same functional group, may show different responses to the plant functional diversity. For example, the predators Chrysopidae and Syrphidae increased with the total amount of available resources [36], but Carabidae also responded to the diversity of these resources [37]. In addition, grasshopper herbivory depended on the plant toughness and insect mandibular strength [28], Carabidae varied in body size and shape with the vegetation structure [38] and Ebeling et al. [22] found that the plant species richness was associated with shifts in many taxa, but not all. Thus, a study about the response of arthropod taxa (and not only functional groups) to the plants might provide further insights.This study only addressed spring months, and arthropod dynamics most probably show seasonal variations.Arthropods were captured using pitfall traps and sweeping nets but different sample methods may capture different arthropods, e.g., [39].We focused on the effects of local factors (i.e., habitats and plants within the habitats), but it is known that the effects of the surrounding landscape structure (i.e., the degree of simplification or heterogeneity/complexity) can dominate the functional community composition and even buffer local effects [40].

## 4. Materials and Methods

### 4.1. Study Sites

Four study areas located in the central part of the PNM were selected. In each area, two habitats with human intervention (one grassland and one chestnut orchard) and two seminatural habitats (one scrubland and one oak forest) were sampled, summing up a total of 16 study sample sites. The minimum distance between study areas was 2 km and the maximum was 8 km. Within the same study area, the minimum distance among the sample sites was 20 m and the maximum 1 km (Figure 5).

The size range of sample sites was between 4400 m^2^ and 10,000 m^2^ for grasslands, 2000 m^2^ and 7000 m^2^ for chestnut orchards, 2500 m^2^ and 20,000 m^2^ for scrublands and 6700 m^2^ and 8400 m^2^ for oak forests. Herbaceous vegetation from grasslands was cut in July for livestock feeding. In the sampled chestnut orchards, the herbaceous vegetation was maintained until November, when the owners cut to facilitate the fruit harvesting. 

### 4.2. Plant Sampling

Plant inventories were carried out within three 28 m^2^ circles in each sample site (around three installed pitfall traps) in May and June 2022. This resulted in a total of 48 plant inventories for characterizing the plant community in each month. The percentage of ground cover for each plant species was recorded following the Daubenmire cover scale modified by Bailey [41] as well as the resource type, the flower color and the plant height.

### 4.3. Arthropod Sampling

Arthropods from the ground and from the vegetation cover were captured in spring (May and June 2022), corresponding with the highest arthropod activity. 

Arthropods from the ground were captured using five pitfall traps in each sample site. Pitfall traps consisted of a plastic cup of a 16 cm depth and 9 cm diameter with a 150 mL mixture of water and polypropylene-glycol (3:1) and 3 or 4 drops of soap which were buried shallow on the surface of the ground. A plastic cover held by three wires, about 3 cm above the surface, was placed over the pitfalls to prevent the entry of small vertebrates and reduce the entry of rainwater (Figure 6). Pitfalls were separated at least 25 m from each other and 25 m from the sample site edge. Thus, a total of 80 pitfalls were sampled per sampling date. Pitfalls were left in the field for 7 days and then collected and stored in the laboratory.

Arthropods from the vegetation cover, including herbaceous, shrubs and tree canopy, were captured using an entomological net. In each sample site, five samples were collected. Each sample consisted of ten sweeps performed in a 10 m transect (a total of 80 samples per sampling date). Each sweep was performed by moving the entomological sweep by 180 degrees. The contents of the net were transferred into a plastic bag and 0.3 mL of diethyl ether were added with a syringe to immediately kill the arthropods.

Arthropods were identified to the maximum taxonomic possible level and further assigned to a functional group (predator, parasitoid, pollinator, phytophagous species, omnivorous, detritivore and parasite).

### 4.4. Plant Traits Selection

The plant traits selection was based on their theoretical meaningfulness for arthropods (trophic interactions and habitat functions), their significance for the occurring plants in the study area (Mediterranean plants) and previous studies of trait-based ecology [1,42,43,44]. The selection also considered the easily quantifiable categorization of the traits through field observations or the literature. Trait information for each plant species was obtained from field observations and from Flora Iberica [45].

For trophic interactions the following functions, traits and categorization were considered:Resource: the resource type at observation (only leaves, leaves and flowers, leaves and fruits.Flower attractiveness:Flower color: brown, pink, purple, red, white, yellow and inconspicuous for not-apparent flowers (e.g., Poaceae, Fagaceae, *Chenopodium* spp. or *Plantago* spp.).Flower area: this was considered as the bloomed area, i.e., the unique flower, the inflorescence or clusters of nearby small solitary flowers and inflorescences forming bloomed clusters. Three categorical levels were established: small = approx. <0.05 cm; medium = approx. 0.05 to 2 cm; large = approx. >2 cm.
Flower accessibility:Flower shape: the flower shape was ranked according to the degree of openness of the corolla in three levels (for each plant species)—total openness (rosaceous and rotate corollas), medium openness (cruciferous, ligulate, hypocrateriform, campanulate and infundibuliform corollas) and low openness (papilionaceus, bilabiate, tubular, personate, orchidaceae and urceolate corollas) (see Aguiar [46] for corolla descriptions and Flora Iberica [45] for species corolla information). This trait was considered as an indication of flower accessibility to arthropods.Nutritional quality:The leaf texture (herbaceous, fleshy, and semi-sclerophyllous/sclerophyllous).The leaf nitrogen (N) and phosphorous (P) composition (mg g^−1^). Mean values—for the species when possible, or for the plant family—from the literature were used [31,32,33,34,35]. For species/families with no information, “Not Available” (NA) was introduced in the data matrix. The values were ranked as follows: for N—low (<19 mg g^−1^), medium (>19 and <23 mg g^−1^), high (>23 mg g^−1^); for P—low (<1.1 mg g^−1^), medium (>1.1 and <1.7 mg g^−1^), high (>1.7 mg g^−1^) (Appendix A). This ranking was arbitrary because there is no previous information about the leaf amount of N and P on the general biodiversity of arthropods. In addition, found interactions related with this trait in this study must be taken as preliminary because plant composition can show a high degree of variability depending on the growing conditions, and there is no information about the composition of these plants in the study area.

For habitat function we considered:Architecture—the plant height at observation (1 = 0 to 5 cm; 2 = 5 to 30 cm; 3 = 30 to 100 cm; 4 > 100 cm). Adapted from Mahdavi and Bergmeier [44] (Table 2).

In all cases, when data were not available, “none” (in the case of leaf N and P and in the case of flower traits when flowers were not present at observation) was used for further calculations.

### 4.5. Plant Functional and Taxonomic Diversity

The plant taxonomic diversity (species richness and Shannon Diversity Index—SDI) was calculated using the *specnumber* function and the diversity function from the “vegan” package [47]. 

The functional diversity for May and June was calculated through a multidimensional functional space construction with a Principal Coordinates Analysis (PCoA) using the package “mDF” [48]. The first four dimensions were used for May and June, since they had the lowest mean absolute deviation (mAD) (mAD May = 0.045; mAD June = 0.044). Then, with the multidimensional space and the plants coverage data, the several functional diversity indexes were calculated [48,49]. Given the absence of studies about this topic in the area all functional indexes calculated by *alpha.fd.multidim* function from the “mDF” package were extracted and further selected (see Section 4.6.2). The final selection (based on the absence on multicollinearity among indexes) was:Functional dispersion: the biomass weighted deviation of species traits values from the center of the functional space filled by the assemblage, i.e., the biomass-weighted mean distance to the biomass-weighted mean trait values of the assemblage. Changes in the functional dispersion reflect changes in the abundance-weighted deviation of species trait values from the center of the functional space filled by the community [49].Functional richness: the volume of multidimensional space occupied by all species in a community within the functional space. The importance of this index relies on the fact that while species (taxonomic) richness is assumed to peak for intermediate disturbance levels, functional richness, through trait reduction, is expected to decrease under high disturbance levels when species decrease [49].Functional divergence (the proportion of total abundance supported by species with the most extreme trait values within a community) and functional evenness (the regularity of the distribution and relative abundance of species in the functional space for a given community): the importance of both traits is based on the fact that after a disturbance, the species abundance is expected to be modified, with species having a combination of traits that are lost and others that remains stable before local extinctions. Thus, reductions in the functional divergence and evenness will reveal disturbance impacts earlier than functional richness [49].Functional specialization: the mean distance of a species from the rest of the species pool in functional space. This indicates generalist species (i.e., species close to the center of the functional space) or specialist species (i.e., having extreme trait combinations) [49].Functional originality: the weighted mean distance to the nearest species from the global species pool. Changes in functional originality quantify how changes in species abundances modify the functional redundancy between species [49].

Then, the species were gathered into functional entities (FE) (i.e., groups of species with the same trait values) using the *sp.to.fe* function from the same package and the functional redundancy and vulnerability calculated:Functional redundancy reflects the average number of species per FE.Functional vulnerability that reflects the proportion of FE with only one species.

### 4.6. Data Analyses

#### 4.6.1. Response of the Plant Taxonomic and Functional Diversity to Habitat and Month

The response of each plant diversity index (taxonomic and functional) to the habitat and month was analyzed using the functions *lm*, *glm* (base R), and *glm.nb* (“MASS” package) [50]. The Poisson (for count data), negative binomial (for over-dispersed count data), and o gamma (for strictly positive continuous data) distributions were used. Overall differences among habitats were checked using the likelihood-ratio Chi-square test with the *Anova* function from the “car” package [51]. A Tukey test for the post hoc analysis was carried out to detect the differences between habitats using the *glht* function from the “multcomp” package [52]. Models were validated using the *simulateResiduals* function from the “DHARMa” package [53].

#### 4.6.2. Response of Arthropod Functional Groups to Plants

The response of arthropod functional groups (abundance of predators, phytophagous species, omnivorous, pollinators, parasitoids and detritivores) to the functional and taxonomic diversity plants indexes was analyzed. The indexes were first standardized and then selected to avoid multicollinearity. Pearson correlations were lower than 0.7 (calculated using the *cor* function from base R). Moreover, a higher variance inflation factor (VIF) than 3 was not allowed, avoiding multicollinearity and minimizing potential model misspecifications [54]. The VIF was 1/(1–ri^2^), where ri^2^ is the determination coefficient of the prediction of all other variables for the ith variable. In our case, values > 3 (ri^2^ > 0.3) indicated variance over 3 times as large as the case for orthogonal predictors [54]. Then, a General Linear Mixed Model (GLMM) was fitted for each response variable (i.e., abundance of predators, phytophagous species, omnivorous, pollinators, parasitoids and detritivores). The selected explanatory variables for the full models were the functional dispersion, evenness, richness, divergence, originality and vulnerability, taxonomic richness, habitat (vegetation or ground) and the month (May or June). The study area was the random factor. The negative binomials (to account for the over-dispersed count data)—lineal (nbinom1) or with quadratic parametrization (nbinom2)—were used. The function *glmmTMB* from the “glmmTMB” package [55] was used to fit the models. The backward selection was performed until all explanatory variables were significant. The most explanatory model (keeping a higher number of explanatory variables) within <2 ΔAIC (Akaike Information Criterion) was selected [56]. Models were validated as in the previous section.

## 5. Conclusions

Regarding the plant functional diversity, in this study some plant traits (namely flower accessibility and color, leaf composition, plant height, leaf consistency or inflorescence area) were relevant to characterize the plant community from a trait-based ecology approach. This information may be important to design future experiments with individual traits. The lower redundancy and higher vulnerability of seminatural habitats indicated that these habitats need special attention to conserve their functional plant diversity. The positive relationship between the plant taxonomic diversity and most of the arthropod functional groups indicated the importance of human intervened areas for the arthropod function during the spring. In relation to the effect of the plant functional diversity, the results (through positive effects of functional dispersion, vulnerability and originality) suggested that the more different and singular the plant traits, the higher the abundance of functional groups. Additional sampling dates in other seasons (summer and autumn) are necessary to understand the whole dynamic over the year. 

## Figures and Tables

**Figure 1 plants-12-00889-f001:**
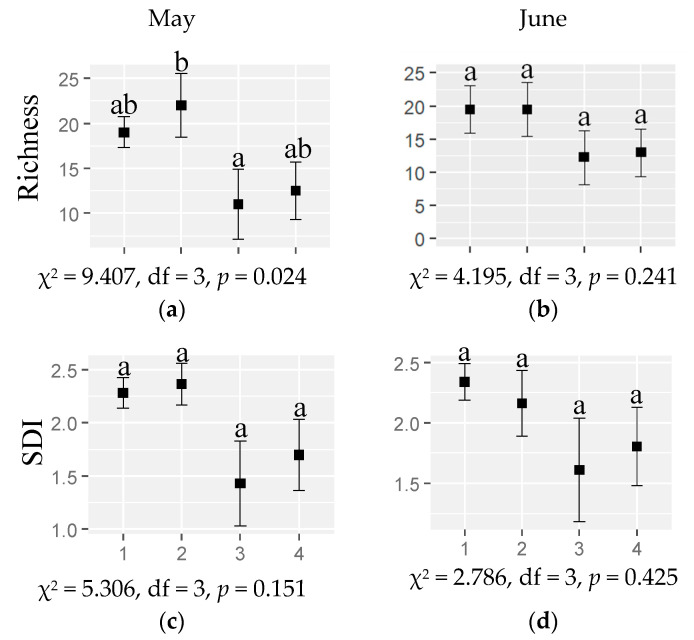
The taxonomic plant diversity: (**a**,**b**) richness (mean ± SE) and (**c**,**d**) Shannon Diversity Index—SDI (mean ± SE) in different habitats (1: grassland; 2: chestnut orchard; 3: scrubland; 4: oak forest) in (**a**,**c**) May and (**b**,**d**) June 2022. Different letters indicate significant differences.

**Figure 2 plants-12-00889-f002:**
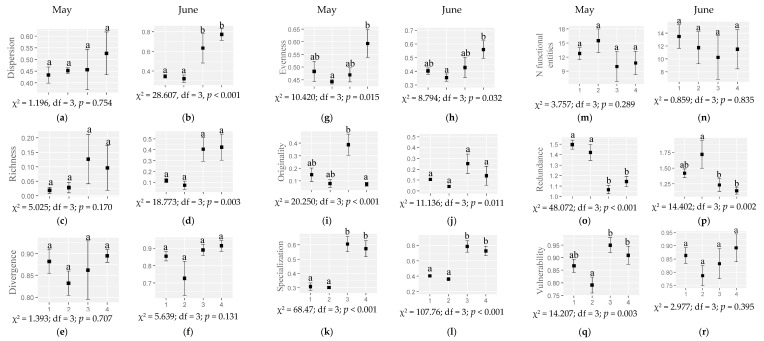
The functional plant (**a**,**b**) dispersion, (**c**,**d**) richness, (**e**,**f**) divergence, (**g**,**h**) evenness, (**i**,**j**) originality, (**k**,**l**) specialization, (**m**,**n**) number of functional entities, (**o**,**p**) redundance and (**q**,**r**) vulnerability (mean ± SE) in different habitats (1: grassland; 2: chestnut orchard; 3: scrubland; 4: oak forest) in May and June 2022. Different letters indicate significant differences among habitats.

**Figure 3 plants-12-00889-f003:**
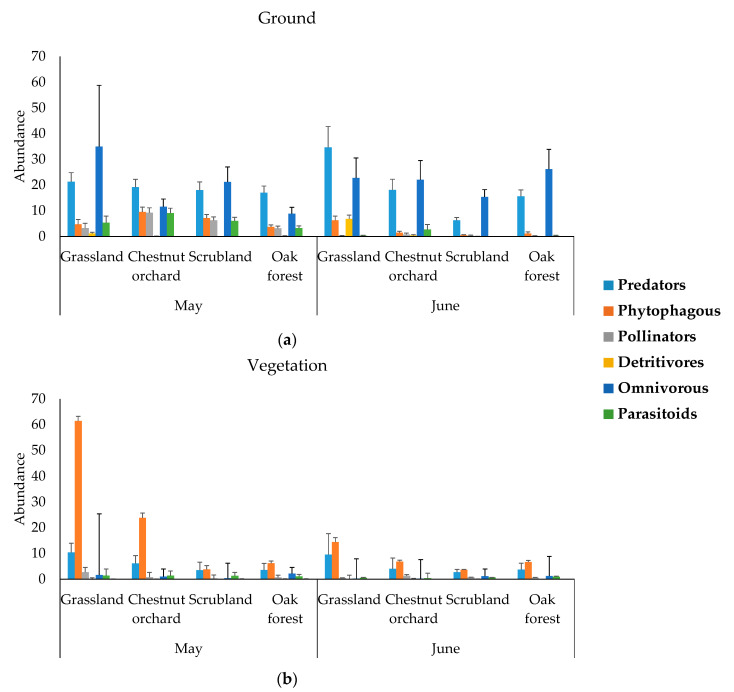
The abundance (mean + SE) of functional groups of arthropods in the (**a**) ground and (**b**) the vegetation.

**Figure 4 plants-12-00889-f004:**
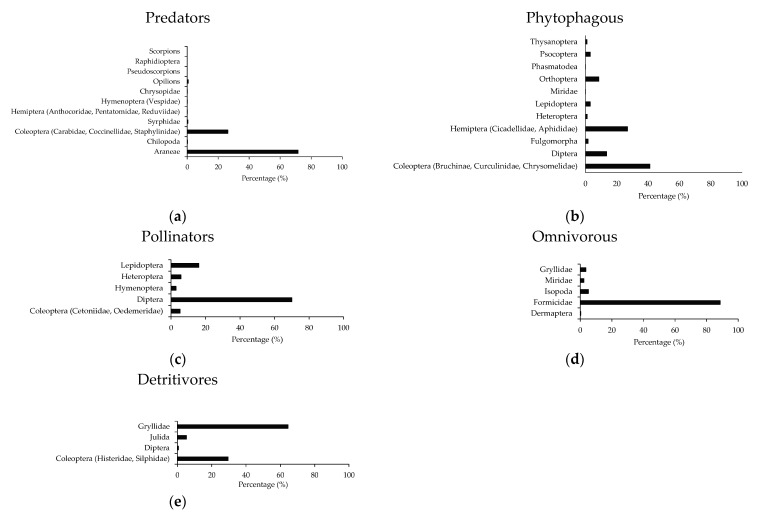
The percentage of the main taxa per functional group of arthropods: (**a**) predators, (**b**) phytophagous, (**c**) pollinators, (**d**) omnivorous, (**e**) detritivores.

**Figure 5 plants-12-00889-f005:**
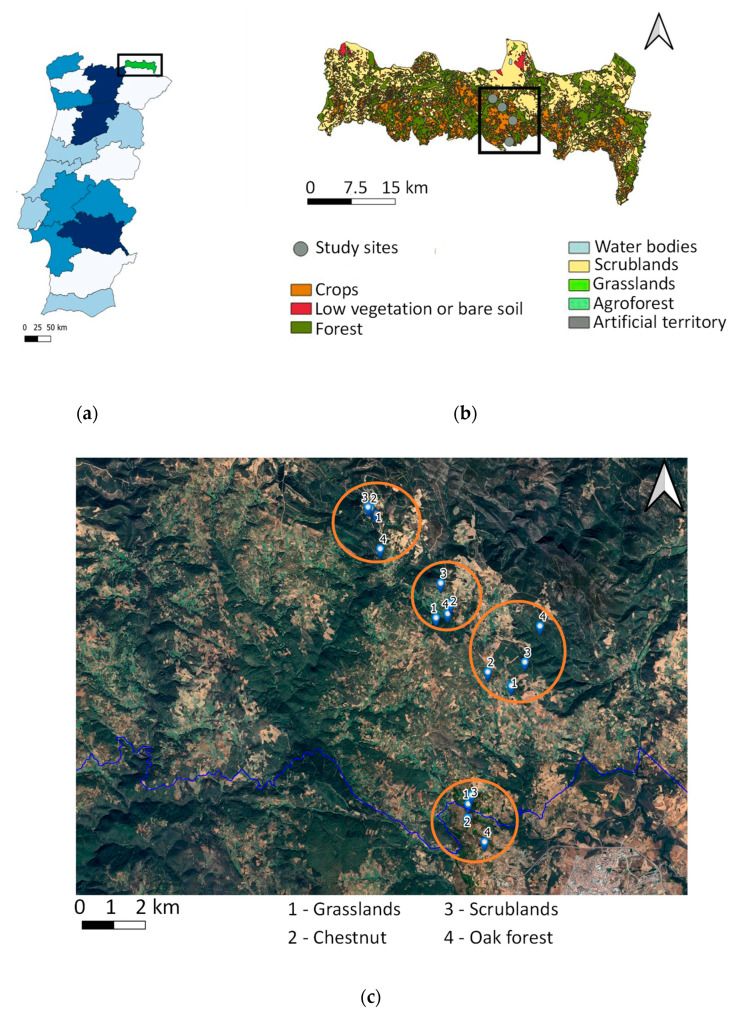
(**a**) The location of the Natural Park of Montesinho (PNM) in Portugal; (**b**) the location of the study areas within the PNM; (**c**) the location of the sample sites within the study area (image obtained from Google Earth^®^).

**Figure 6 plants-12-00889-f006:**
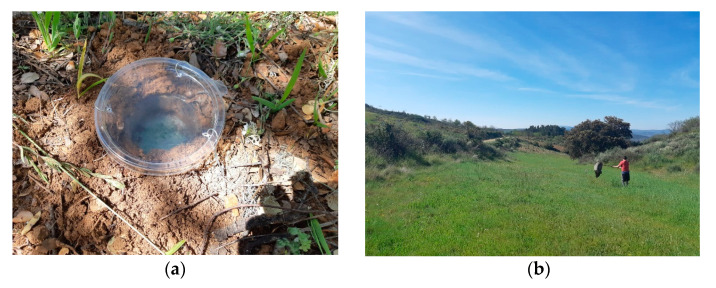
(**a**) An illustration of a pitfall trap; (**b**) an illustration of the sweeping net sampling.

**Table 1 plants-12-00889-t001:** The GLMM results for the response of predators, phytophagous, pollinators, omnivorous, parasitoids and detritivores to the plant functional and taxonomic diversity, the sampled habitat (ground and vegetation) and the month (May and June).

Predators					Phytophagous				
	β	SE	Z	*p*		β	SE	Z	*p*
(Intercept)	3.102	0.121	25.602	<0.001	(Intercept)	1.100	0.178	6.194	<0.001
Functional dispersion	0.214	0.100	2.142	0.032	Functional dispersion	0.507	0.149	3.409	0.001
Functional richness	−0.536	0.116	−4.625	<0.001	Functional evenness	−0.235	0.102	−2.306	0.021
Functional vulnerability	0.197	0.062	3.156	0.002	Functional richness	−0.987	0.185	−5.334	<0.001
Taxonomic richness	0.230	0.070	3.308	0.001	Functional originality	0.227	0.106	2.141	0.032
June (vs. May)	−0.382	0.139	−2.744	0.006	Functional vulnerability	0.130	0.103	1.266	0.205
Vegetation (vs. Ground)	−1.248	0.102	−12.200	<0.001	Taxonomic richness	0.416	0.109	3.802	<0.001
					June (vs. May)	0.379	0.228	1.662	0.097
					Vegetation (vs. Ground)	1.214	0.155	7.824	<0.001
**Pollinators**					**Omnivores**				
	β	SE	Z	*p*		β	SE	Z	*p*
(Intercept)	−0.377	0.215	−1.752	0.080	(Intercept)	2.892	0.216	13.359	<0.001
Functional dispersion	0.196	0.116	1.696	0.090	Functional evenness	0.162	0.104	1.550	0.121
Functional evenness	−0.369	0.120	−3.066	0.002	Functional richness	−0.052	0.095	−0.552	0.581
June (vs. May)	1.872	0.234	8.016	<0.001	Functional divergence	0.095	0.106	0.896	0.370
Vegetation (vs. Ground)	−0.687	0.209	−3.295	0.001	Functional originality	0.176	0.107	1.639	0.101
					Taxonomic richness	0.285	0.123	2.324	0.020
					Vegetation (vs. Ground)	−3.085	0.185	−16.704	<0.001
**Parasitoids**					**Detritivores**				
	β	SE	Z	*p*		β	SE	Z	*p*
(Intercept)	−0.007	0.238	−0.031	0.975	(Intercept)	0.009	0.326	0.028	0.977
Functional dispersion	0.177	0.172	1.029	0.304	Functional richness	−1.438	0.432	−3.328	0.001
Functional evenness	−0.187	0.112	−1.680	0.093	Functional vulnerability	0.650	0.255	2.547	0.011
Functional richness	−0.112	0.203	−0.554	0.580	Taxonomic richness	0.486	0.190	2.550	0.011
Functional originality	0.115	0.092	1.251	0.211	June (vs. May)	−2.305	0.516	−4.470	<0.001
Taxonomic richness	0.071	0.100	0.712	0.477					
June (vs. May)	1.530	0.297	5.153	<0.001					
Vegetation (vs. Ground)	−0.619	0.175	−3.534	<0.001					

**Table 2 plants-12-00889-t002:** The ecosystem function for arthropods’ functional traits, variable category, classification and source of the trait selection and trait level classification [1,29,31,32,33,34,35,44,45,46].

Ecosystem Function for Arthropods	Functional Trait	Variable Category	Classification	Trait Selection Source	Level Classification Source
Resource	Resource type	Nominal	1 = Vegetative (leaves); 2 = Flower (leaves and flowers); 3 = Fruit (leaves and fruits)	Field observation	Field observation
Attractiveness	Flower color	Nominal	1 = brown; 2 = inconspicuous; 3 = pink; 4 = purple; 5 = red; 6 = white; 7 = yellow	Field observation	Field observation
	Flower area	Ordinal	small = approx. < 0.05 cm; medium = approx. 0.05 to 4 cm; large = approx. > 4 cm	Adapted from Fornoff et al., [29]	Classified according to potential arthropod meaningfulness
Accessibility	Corolla shape	Ordinal	1 = total openness; 2 = medium openness; 3 = low openness	Adapted from Fornoff et al., [29]. Flowers typologies from Aguiar [46]	Classified according to potential arthropod meaningfulness
Nutritional quality	Leaf texture	Ordinal	1 = herbaceous; 2 = fleshy; 3 = Semi-sclerophyllous or sclerophyllous	Leaves typologies by consistency from Aguiar [46]	Literature
	P	Ordinal		Literature	Literature
	N	Ordinal		Literature	Literature
Architecture	Plant height at observation	Ordinal	1 = 0 to 5 cm; 2 = 5 to 30 cm; 3 = 30 to 100 cm; 4 = > 100 cm	Adapted from Mahdavi and Bergmeier [44]	Observation

## Data Availability

The data that support the findings of this study are available from the corresponding author upon reasonable request.

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
