# Peer review of "Plant Functional Dispersion, Vulnerability and Originality Increase Arthropod Functions from a Protected Mountain Mediterranean Area in Spring"

_plants, 2023, doi:10.3390/plants12040889_

Round 1

Reviewer 1 Report

In this manuscript, the authors sampled the plant and arthropod communities in four habitats (chestnut orchards, grasslands, scrublands, and oak forests) in the Natural Park of Montesinho (NE Portugal) with the intention of assessing the relationship between plant and arthropod communities through an analysis of the plant functional traits. Overall, the authors have done numerous and complex analyses and I am a bit surprised that the main conclusions are that all four habitats contribute to the taxonomic and functional diversity of the plant community (L237-239) and that plant taxonomic diversity supports arthropod functional richness (L245-246). The conclusions associated to the GLMs with the arthropod functional groups are extensively discussed despite the issues with the models, and I am not convinced these results are reliable (hopefully the random structure would solve these issues). Below, I provide some major comments that, in my opinion, need to be addressed to make this work publishable.

Introduction

The only problem I see is the long and, in my view, mostly irrelevant paragraph L58-74. Here the authors have mixed several heterogeneous studies just to show that other research have used the same functional ecology approach. However, it is unclear whether the examples mentioned are any relevant for this particular study. The introduction is structure as follows:

1.Importance of plants for arthropods,

2.Importance of plant functional traits for arthropods,

3.L58-74 seems like a different paragraph, not well connected to the previous ones, and more in the spirit of discussion rather than introduction (the authors started this paragraph mentioning the challenges of this approach, which could also sound counterproductive). I don’t think that justifications for the use of a functional ecology approach are needed nowadays, and if the authors want to highlight limitations and or compare their results with other *relevant* studies that used the same approach, they should do it in the discussion.

4.The study context

5.Aim. Here it could also be of additional value to state some expectations even if it is clear that the study was explorative.

Materials and methods.

The two major problems with this section are 1) missing or unclear information and 2) the GLMs.

There is some confusion with the naming of the subplots, which I think are the 28 m2 circles.

Some clarification about the kind of vegetation sampled in the four study areas/plots using sweeping sampling is needed. Did you sample only herbaceous species, also shrubs or tree canopy?

In the Results, you use adjective such as “relevant” and “most abundant” and it is unclear what you are presenting here (also because there are no numbers shown).

Plant sampling: the inventories were performed in 3 occasions between May and June, but then you mentioned that the two months were considered separately when looking at the traits (L468-ff). Was month a factor to be explored? (if so possibly more temporal replicated were needed) Would the fact that one of the two months was sampled twice be problematic?

I am a bit skeptical of the usefulness of nutrition quality as a trait taken from the literature. Frankly, it is not really very convincing, but the authors recognise it. Have you consider to drop it?

Concerning the GLMs, within each study area, there were 4 habitat that are nested. These 4 habitats may be more similar to each other than other far away because of their closeness. A GLMM would have been more appropriate.

Results

In general, the Results section is very heavy because the authors tested for many things. It is difficult for the reader to distinguish what are the most important results. Moreover, some are presented in a list form which is unusual for text (perhaps put these values in a table instead). Pie charts are also not very informative (% of less represented groups are impossible to be read).

I wonder whether the model misfits are due to the lack of the random term (the plots should nested within the study area to take into account also that covariance of the pitfall trap within the same plot) and also how did the validation plots looked. Perhaps some insight that could help to improve the models could be obtained by plotting the model residuals vs. every factor in and out of the model. I don’t think that presenting the output of an unreliable model makes sense, and if the issues cannot be solved, perhaps it should be discarded.

Discussion

L240-243 What are the conclusions then? That these traits could be used instead of others for this kind of analyses? Or that these particular traits are what makes the plant community so rich in functions? I can imagine that these traits would have an impact on the arthropod community, but it is not clear why it is important in themselves.

Conclusions

L536-537 I wonder if this inconclusiveness can be due to the fact that the authors took presumably the traits of the adult arthropods, which can be totally different from that of their larval stages. In general, L537-ff is not very convincing and it is just an articulate way to say that more studies are needed. A better conclusion should be found.

The manuscript requires the meticulous revision by a native English speaker or a professional Editor. There are numerous sentences that are unclear or incorrect. I have spotted several and provided suggestions in the minor comments below, however, I have surely missed some.

Minor comments:

As it is, the title misses the opportunity to summarise the most important results from this study. I recommend the authors to consider also alternative, more informative, titles. This is merely a friendly suggestion rather than a critic and if they prefer a more general title like the current one this is not a problem. 

The abstract is overall OK, but it is unclear which traits are important, L21 just mentions “single traits”.

The keywords are good.

Abstract

L13 our goal was.

Introduction

L30 oviposition or mating places (without comma)

L30-31 I think it would sound better (and be more correct) if you change “or” to “and” in both sentences.

L31 change “for the” to “for several”.

L36 ecosystem functions

L37 arthropod biodiversity.

L38 comma after “climate change”

L40 establish

L44 focuses

L47 functions

L51 For example,

L55 characteristics

L58-59 “The functional ecology also implies several challenges”. This sentence is unclear also because it is not well connected to the previous paragraph. It seems to start a different topic and some better articulation.

L67 a functional ecology approach can reveal

L67 remove “very” and perhaps “interesting” is not the best adjective here.

L68-70, I would remove the details in parenthesis and rephrase “arthropod (?) herbivores are more affected/depend on food quality, while arthropod (?) predators by habitat volume”. However, in this sentence there are many unclear aspects.

Did this study focus on arthropod herbivores and predators or other groups?

How should we interpret habitat volume? It is a measure of habitat amount and quality (tridimensionality of the pasture?)

The original text reads “herbivore track food quality”, but “track” does not seem the correct verb here. It is unclear what is the response affected by food quality. Is it arthropod abundance, species richness, diversity or something else?

L76-79 This sentence is not really informative. It just states that not many arthropod studies took place in the area. It would be much more important to provide some detail about the main habitats and perhaps the arthropod communities in this region.

Material and methods

L372 How distant from each other were those study areas?

L383-384 I think this information was already given in the introduction.

L389-390 in May and June 2022 within 28 m2 circles in each habitat. Perhaps in this way you can avoid repeating “plot”, which is confusing since you have 4 study areas each with 4 plots and then 3 plant inventories based on 28 m2 circles (instead of plot again).

Table 2. I suggest you use always English (‘red’ instead of vermelho, ‘white’ for branco and ‘yellow’ for amarelo). Space missing between ‘30’ and ‘cm’.

L400 remove “in each subplot” you mention it twice in the sentence.

L401 where are the ‘subplots’? I guess these were the 28m2, but the authors should use the names consistently for clarity.

L403 no comma before ‘and’ and 25 m.

L406 which vegetation cover were considered in these 10 m transects? It is unclear.

L433 established.

L442 why “Semi-…” capitalised?

L444 Nitrogen (N)

L447 was introduced. However, this sentence needs complete rewriting.

L449, 450 comma before “high”

L458 30 cm

Results

L92-93 do not connect well (it is grammatically incorrect) with what comments after the colon.

L95-ff what does “relevant” species mean? Were those dominant? Unique of this habitat? Something else? This adjective is misused also in other sentences.

L105 what do you mean with “most abundant”? They occupied higher % of ground cover?

L179 captured

L199 Table

L202 remove this sentence about model validation. This was mentioned in M&M.

Discussion

L232 chestnut orchards

L253-254 The way I see it, the authors did not specifically test for the effectiveness of management strategies in the region and should avoid making conclusion about that.

L258 would it better to use “unique traits” instead of “single traits”? The latter gives the impression that it is only 1 trait consider at the time that is important.

L268 establish

L271-272 “the behaviour of the sampled arthropod community “. I understand what you mean but it does not sound correct in English. Please rephrase.

L272-273 “i.e., the traits might not perform the expected ecosystem 272 functions for the arthropods support.” Unclear.

L368 inconclusive

Conclusions

L528 to describe the arthropod community? 

Reviewer 2 Report

The paper is interesting and original. As I wrote in the original text, regrettably the monitoring period was limited to spring months (and for this reason, I suggest changing the title), while in the Portuguese monitored area the months July-September are certainly rich in insects, particularly species living on flowers or also on the ground. The pitfall traps may intercept a high number of ground-dweller species, but not all (e.g.: they capture Gryllidae but not Acridoidea). I suggest highlighting the existence of limits in the method of capture.

Finally, even if monitoring was carried out in May and June (line 399), in the results the authors discuss also of July month. Please, check carefully.

Reviewer 3 Report

After reading the manuscript, I think that the topic of the MS is suitable for the journal and will be relevant to an audience of researchers. This study provides new and valuable data about the local/regional relationships between the two major groups of organisms in terrestrial ecosystems, in terms of diversity and biomass. In general, I like this manuscript.

The English text is understandable, and it is in general well-written, but it could and should be improved (grammar, syntax, structure of sentences, odd phrasing, etc.). I recommend you seek the assistance of a native English speaker or an expert linguist, preferably one who has some knowledge of botany/ecology.

You need to integrate the statistical data obtained in the results in the text of the discussion, many sentences of the discussion have to be defended with the numerical data obtained, this would greatly improve the discussion.

The sample size is fine, and statistics seem fine to me.

Other issues:

Line 131: add space after (d).

Figure caption of Fig. 1 must be explained in more detail, hard to follow.

Linbe 240-241: please, explain that in more detail.

Lines 245-246: "The arthropod functional richness increased with the plant taxonomic richness (higher in chestnuts orchards)." I agree, but you should support the sentence with the statistical data, please add the data that evidence that in the discussion.

Lines 306-310: "Pollinators and parasitoids were more abundant in May than in July and this may be related with a higher resource amount, although the month did not affect the other groups". This could be true for pollinators for ovious reasons, but Im not sure about parasitoids, please explain in detail why in May parasitoids habe a higher resource amount.

Lines 347-355: Why do you think that you have not found effect of plant diversity on phytophagous or parasitoids, this is interesting and it need a better explanation.

Reviewer 4 Report

The manuscript “Bottom-up relationships between the plant and the arthropod communities in a protected mountain Mediterranean Area” describes the vegetation and arthropod diversity and the influence of vegetation on arthropod diversity in different habitats of the Montesinho Natural Park.  The manuscript includes a lot of data gathered in May and June 2022, but some parts of the methods and analysis are confusing and require clarification.  Perhaps some of the functional diversity indexes and some of the functional traits could be removed.

Results

Although some bullet points can be included in each subsection, it would be better to at least introduce the subsection before using bullet points.

Pages 2-3, lines 95, 100, etc.: What do you mean by “some relevant species”? were these the most abundant?

Page 3, line 113: remove the accent from angustifolia.

Page 3, lines 117-122: what is the biological relevance of the different PCs?  This should be further explained.  For example, when the authors say that leaf P contributed for PC 1, 2 and 3, what is the ecological significance of this?

Pages 4-5: Additional information regarding why these functional diversity indexes were used should be added to the Introduction and/or to the Material and Methods section.

Page 6, subsection 2.5, figure 3, and page 8, lines 213-215:  Data from pan traps can be misleading as many insects passing by drawn.  It does not make sense that pollinators were more common on the ground than on vegetation and this seems to be a limitation of the methodology used.  Pollinators may be just passing by while moving between plants or moving to different habitats.  Visual counts would have provided more reliable data on the association between flowering plants and pollinators.

Materials and Methods

Pages 11-12, section 4.1: Regarding the study sites, the use of the term “plot” is sometimes confusing.  Rather than “study plots”, the authors should use “study sites”.

Page 13, line 389: why was the analysis divided for May and June? why did the authors did not combine those data to simplify the presentation of the manuscript?

Page 13, line 400: I think these are not pitfall traps, these are pan traps

Pages 13-14, Table 2, and lines 444-456.  The values and references used for N and P values should be included in the manuscript as a table in the supplementary material.  N and P values can change with plant growth and depend also on the soil where plants are grown.  Thus, different studies may provide different values for the same plant species.  How did the authors address this variability? The authors mention some of this limitation in lines 543-456 and later in the Discussion in lines  273-275, but the information used should be provided, so it is more clear how this limitation was addressed. 

Page 14, line 435-440.  Rather than referring to another publication, further description of the methodology used to assess accessibility should be included.  Furthermore, when comparing plants within the same family, some species have flowers that have with much larger corollas than others.

Page 15, lines 470-495.  The authors should explain why these indexes were selected and furthermore information on how these indexes were calculated should be added.  I would recommend reducing the number of indexes.  Some of them also seem to be correlated, for example redundancy and overredundancy.

Page 16, line 515, explain what the variance inflation factor is.

Figures S2 and S3: the authors should provide more information on how the interpretation of these figures has been done, indicating also their meaning in terms of biological relevance for arthropods.

Round 2

Reviewer 1 Report

I thank the authors for their revision. I reckon that the manuscript has improved. I have a few more questions/comments.

When the authors mentioned missing values for some of the variables, they should specify their %s (e.g., L370, but also for other traits). This is because missing values are critical in functional trait analysis and based on how they are handled, the results may change. When functional diversity is calculated, it has to be clarified whether rows with NA value were discarded or alternatively, how they were treated.

In the reply letter, the authors mentioned that the models for species richness and SDI were removed, but taxonomic diversity indices are mentioned at L111-113, L388-390, and L426.

L144 Were parasitoids more abundant than omnivores? Fig.3a looks highly dominated by predators and omnivores. Moreover, I am afraid that Fig.3a and 3b were switched. Typically, pitfall traps collect large numbers of predators and omnivores and sweep netting, high numbers of herbivores (see also L158).

L23 color (US spelling as you used elsewhere).

L461 Full stop missing.

Author Response

We are very grateful for reviewer#1´s comments and suggestions which clearly helped us to improve the manuscript.  

Responses to Reviewer#1

I thank the authors for their revision. I reckon that the manuscript has improved. I have a few more questions/comments.

When the authors mentioned missing values for some of the variables, they should specify their %s (e.g., L370, but also for other traits). This is because missing values are critical in functional trait analysis and based on how they are handled, the results may change. When functional diversity is calculated, it has to be clarified whether rows with NA value were discarded or alternatively, how they were treated.

In all cases, when data were not available “none”, as an additional level, was used for further calculations. This information was added in lines 400-402. Please, see also Figure S2 and S3 for the values of “none” in each variable (in Leaf-P and Leaf-N, or in flower characteristics when the plant species did not present flowers at observation).

In the reply letter, the authors mentioned that the models for species richness and SDI were removed, but taxonomic diversity indices are mentioned at L111-113, L388-390, and L426.

The models in the mentioned lines were not removed. Please, notice that the removed models were relative to arthropods and the mentioned lines are models for plants, i.e., the removed models were about the effect of plant variables on arthropods richness and SDI (Line 191 to 215, 468-469, in the original manuscript). The lines mentioned are plant richness and SDI across habitats in the different dates. Later (plant) models were validated and not removed.

L144 Were parasitoids more abundant than omnivores? Fig.3a looks highly dominated by predators and omnivores.

The sentence was rephrased:

“Predators and omnivorous were the most represented functional groups in the ground while phytophagous and predators were the most abundant groups in the vegetation” Please, see lines 150-151.

 Moreover, I am afraid that Fig.3a and 3b were switched. Typically, pitfall traps collect large numbers of predators and omnivores and sweep netting, high numbers of herbivores (see also L158).

Caption had a mistake. We corrected it. Please, see line 155-156.

L23 color (US spelling as you used elsewhere).

Color was corrected (also in other parts of the manuscript) (line 23, 196, 235, 274, 326, 364, 492, table 2)

L461 Full stop missing.

Full stop was added (please, see line 495).

Reviewer 4 Report

The manuscript “Bottom-up relationships between the plant and the arthropod communities in a protected mountain Mediterranean Area” has been improved after the revisions, but additional changes are necessary because not all my concerns have been completely addressed. 

Pages 3-4, lines 188-197: further explanation is necessary to explain the biological relevance of the different PCs.  For example, when the authors say that leaf P contributed for PC 1, 2 and 3, what is the ecological significance of this? How different is the significance of the different traits depending on whether they contributed to a certain number of principal components of the PCoA (for example two or three)?

Pages 9-10, lines 396-422.  As I recommended before, additional information regarding why these functional diversity indexes were used should be added to the Introduction and/or to the Material and Methods section.  The authors should explain why these indexes were selected.  Some of the indexes, such as “functional divergence”, “functional mean pairwise distance”, and “functional overredundancy”, do not seem to bring anything meaningful to the study and I would recommend deleting them.

On page 5, line 250, replace “visual counts for pollinators” by “visual counts of pollinators visiting flowers”.  As I mentioned earlier, data from these traps can be misleading as many insects passing by drawn on the water.  It does not make sense that pollinators were more common on the ground than on vegetation.  Pollinators passing by while searching for flowering plants draw on the traps.  This seems to be a limitation of the methodology used. Counts of pollinator visits to flowering plants would have provided reliable data on the association between flowering plant diversity and pollinators.

On line 260: delete “probably”, they for sure are variable

Page 8, lines 359-364.  As the publication by Aguiar provides only corolla descriptions and does not provide information on pollinator accessibility to corollas, where did you get the information on plant families and accessibility from?  Supporting references should be provided.  Furthermore, when comparing plants within the same family, there is some variability and some species have flowers with much larger corollas than others.

More information should be added on the interpretation of figures S2 and S3.  What is written on lines 185-186 was already there before the revision.

Author Response

We are very grateful for reviewer#4´s comments and suggestions which clearly helped us to improve the manuscript. We tried to improve clarifications to reviewer#4’s concerns, hoping that they are now clearer. Thank you for the effort.

Responses to Reviewer#4

Reviewer#4

The manuscript “Bottom-up relationships between the plant and the arthropod communities in a protected mountain Mediterranean Area” has been improved after the revisions, but additional changes are necessary because not all my concerns have been completely addressed. 

Pages 3-4, lines 188-197: further explanation is necessary to explain the biological relevance of the different PCs.  For example, when the authors say that leaf P contributed for PC 1, 2 and 3, what is the ecological significance of this? How different is the significance of the different traits depending on whether they contributed to a certain number of principal components of the PCoA (for example two or three)?

We tried to explain better/clarify: this result does not have a biological meaning itself, but it rather constitutes relevant information for further research. For example, if we choose the trait “flower color” because it is generally important to many arthropods, but all flowers are yellow, this trait will not contribute to the variability of the plant community and it will not be worthy to further explore potential relationships with arthropods. In this example, it is clear that flower color would not have any contribution to further analysis, but in our study we used the multivariant analysis to elucidate if our selected traits vary across the vegetal community or not. Hoping that it helps, we tried to explain as follows:

“In this study, the PCoA showed the relationships among the plant traits within the plant community.  Results indicated that the flower accessibility, the flower color, and the leaf N composition were relevant traits to characterize the plant functional diversity in the PNM (i.e., they significantly contributed to the trait variability within the plant community). Other important traits were the leaf P composition, the plant height, the leave consistency, and the inflorescence area (Figure S1 and S2 shows the significant contribution of each trait to each PCoA axis in May and June). The variability of these traits across the plant community indicates them as suitable candidates to investigate arthropod-plant interactions in further research in the PNM (i.e., a plant trait generally important for arthropods can be selected, but if this trait does not vary across the plant community, it will be meaningless for the arthropod community). We recall that leaf N and P composition were extracted from literature, therefore this result must be considered as exploratory and further research performed for verification.” Please, see lines 195-209.

Pages 9-10, lines 396-422.  As I recommended before, additional information regarding why these functional diversity indexes were used should be added to the Introduction and/or to the Material and Methods section.  The authors should explain why these indexes were selected.  Some of the indexes, such as “functional divergence”, “functional mean pairwise distance”, and “functional overredundancy”, do not seem to bring anything meaningful to the study and I would recommend deleting them.

We followed the reviewer suggestion and deleted “functional mean pairwise distance”, and “functional overredundancy” Funcional divergence was maintained because it was included in the full models and - despite with no significant effect on the response - it is within the explanatory variables of the selected model for Omnivores.

We gave additional details about the functional indices selection and their meaning. We extracted all the calculated functional diversity indexes by mDF (based on the absence of previous knowledge) and then selected them accordingly to the absence of multicollinearity. We tried to clarify this (please see lines 423-426). We included additional information about the functional indices meaning in the material and method section to clarify the relevance of the used indices (please, see lines 427-453).

On page 5, line 250, replace “visual counts for pollinators” by “visual counts of pollinators visiting flowers”.

We followed the reviewer suggestion and “visual counts of pollinators visiting flowers” was included. Please, see lines 262-263.

  As I mentioned earlier, data from these traps can be misleading as many insects passing by drawn on the water.  It does not make sense that pollinators were more common on the ground than on vegetation.  Pollinators passing by while searching for flowering plants draw on the traps.  This seems to be a limitation of the methodology used. Counts of pollinator visits to flowering plants would have provided reliable data on the association between flowering plant diversity and pollinators.

We followed the reviewer and we included “visual counts for pollinators visiting flowers, since pollinators in pitfalls can draw on pitfall traps while searching for flowering plants” (Please see line 262-264) (however, the pitfall plastic cover probably also prevent, at least in some extent, flying insects drawn). Also, please see line 254, after the previous review, results modified and both phytophagous and pollinators were statistically more abundant in the vegetation (fortunately this makes more sense), although this is not occurring for parasitoids (still difficult to explain and the capture method can be an explanation). The capture method limitation is in line 297-298).

On line 260: delete “probably”, they for sure are variable

Following the reviewer suggestion, “Probably” was replace by “must”. Please, see line 273.

Page 8, lines 359-364.  As the publication by Aguiar provides only corolla descriptions and does not provide information on pollinator accessibility to corollas, where did you get the information on plant families and accessibility from?  Supporting references should be provided.  Furthermore, when comparing plants within the same family, there is some variability and some species have flowers with much larger corollas than others.

Information about plant traits (including corolla shape) for each species was obtained from Flora Iberica [reference 46 in the manuscript]. (Please visit http://www.floraiberica.es/eng/index.php). This is a project which synthesize current knowledge about vascular plants which spontaneously grow in Iberian Peninsula and Balearic Islands. Traits were described by species (not by family). We tried to clarify in the manuscript (please, see lines 257).

Following the reviewer concern, we change the name of the trait from “accessibility” to “flower shape” (all across the manuscript, including supplementary material). Despite being an indicator of accessibility to arthropods, with the flower morphology we did not intend to indicate a confirmed arthropod accessibility to flowering species by previous studies, but just grouping the flowers in a wide rank of flower openness. We hope is now clearer (please, see lines 372-380).

More information should be added on the interpretation of figures S2 and S3.  What is written on lines 185-186 was already there before the revision.

Please, see first comment.

Round 3

Reviewer 4 Report

The manuscript has been improved with the changes done by the authors and now it can be accepted for publication.